# Microbial Exopolysaccharides as Drug Carriers

**DOI:** 10.3390/polym12092142

**Published:** 2020-09-19

**Authors:** Antonio Tabernero, Stefano Cardea

**Affiliations:** 1Department of Chemical Engineering, University of Salamanca, Plaza los Caídos s/n, 37008 Salamanca, Spain; antaber@usal.es; 2Department of Industrial Engineering, University of Salerno, Via Giovanni Paolo II, 132, 84084 Fisciano, Italy

**Keywords:** exopolysaccharides, biomaterials, drug delivery systems

## Abstract

Microbial exopolysaccharides are peculiar polymers that are produced by living organisms and protect them against environmental factors. These polymers are industrially recovered from the medium culture after performing a fermentative process. These materials are biocompatible and biodegradable, possessing specific and beneficial properties for biomedical drug delivery systems. They can have antitumor activity, they can produce hydrogels with different characteristics due to their molecular structure and functional groups, and they can even produce nanoparticles via a self-assembly phenomenon. This review studies the potential use of exopolysaccharides as carriers for drug delivery systems, covering their versatility and their vast possibilities to produce particles, fibers, scaffolds, hydrogels, and aerogels with different strategies and methodologies. Moreover, the main properties of exopolysaccharides are explained, providing information to achieve an adequate carrier selection depending on the final application.

## 1. Introduction

Over the last decades, pharmaceutical, biotechnological, and medical companies have tried to develop administration systems to improve drug therapeutic effects and their half-life without producing side effects. For this reason, drug delivery systems (DDSs) have been designed to keep blood drug concentrations from fluctuating, and to reduce dosage frequency. However, in order to design a DDS efficiently, the complete knowledge of a drug’s properties, system biocompatibility, and administration route is previously required [1]. 

One of the most important parameters for a DDS is its size because it is a pivotal factor for targeting the organs. Pulmonary delivery requires microparticles (1–5 microns) to assure a correct deposition in alveoli and airways [2,3]. Particles 10–600 nm are required for transdermal delivery to achieve a correct drug delivery into the skin’s layers. In addition, nanometric particles are required to improve the possibilities of overcoming the blood-brain barrier in brain delivery systems. Likewise, nasal delivery requires particles ranging from 8 to 20 microns, whereas particles between 200 nm and 2 microns are required for intravenous administration [2,3].

DDSs are usually composed by a carrier and the drug to be delivered. The carrier is usually a biodegradable and biocompatible polymer that protects the drug from degradation in the human body. The selection of the carrier is also a key parameter for controlling the subsequent release mechanism: the drug diffuses from the polymer and/or the polymer swells to improve the drug’s release, or the drug can be released due to a dual mechanism erosion-diffusion. Surface properties are other parameters that have to be considered when selecting the proper carrier since they play an important role in blood and tissues compatibility [1,2,3].

Different polymers have been used as carriers. Lactic acid derivatives and its copolymers, such as poly (l-lactic acid) (PLLA) or poly (lactic-co-glycolic acid) (PLGA), have been widely considered by many researchers [4,5,6]. Another interesting approach is the use of free-radical scavenging materials, mimicking melanin, such as polydopamine, that can generate oxidative stress to treat different diseases [7]. Additionally, polydopamine has an affinity for various ions, and as a consequence, some dyes can be even incorporated in this polymer for medical imaging applications [8]. The concept of “click chemistry” can be also considered to modify and generate new polymeric biomaterials, with any type of property, for different applications, such as tissue engineering [9].

Another option is the use of polysaccharides [10], which are natural compounds found in plant cell walls (cellulose) and also used as energy storage units (starch). Basically, these polymers are chains of carbohydrate molecules that usually are linked by glycosidic linkages. Polysaccharides are classified depending on how many types of monosaccharides compose the polymer (homopolysaccharide or heteropolysaccharide) or depending on their biochemical origin [10]. 

Among the polysaccharides, it is possible to find a special type called exopolysaccharides (EPSs), which can be produced by bacteria or other organisms. Microorganisms require these compounds to assure their structural integrity [11,12]. 

EPSs are usually excreted into the environment from different organisms, and consequently they can be extracted from the medium culture if the organisms are cultivated with fermentative processes. EPS formation can be explained by different mechanisms, such as synthase-dependent pathways, Wxz or Wzy-dependent pathway, or even because the bacteria or the organisms (i.e., red seaweed) may secrete an enzyme that polymerizes the required sugar source, forming the polymer in the medium culture (extracellular synthesis with enzymes). More information about these pathways can be read elsewhere [12,13,14].

Therefore, their industrial production is based on cultivating the bacteria (or another living organism) with a subsequent centrifugation to remove the biomass. After that, an extractor agent is added to the supernatant to precipitate the polymer. The last steps are polymer centrifugation and purification. A similar process has been followed to produce EPSs, such as alginate [15], dextran [16], xanthan gum [17], gellan gum [18], curdlan [19], levan [20], hyaluronic acid [21], pullulan [22], carrageenan [23], etc.

However, another methodology has been proposed in which some EPSs can be obtained, if the EPS follows an enzymatic extracellular synthesis, by using the purified enzyme with a solution of the sugar, avoiding the use of bacteria (cell-free system) [24]. Although this technique is faster, the enzyme has to be previously obtained or acquired from the market.

In any case, EPS production results easier than the production of other intrapolysaccharides, in which additional steps are required in downstreaming and purification processes. 

EPSs also have suitable properties for the biomedical industry. In addition to having biocompatible and biodegradable properties, they are highly hydrophilic and usually form pseudoplastic solutions in water. These rheological characteristics make them good candidates for applications in which viscosity can play an important role, such as cosmetics or food. Furthermore, due to their polymeric structure, hydrogels can be produced from EPSs by using a crosslinker or physical methodologies [25,26,27]. 

The potential of exopolysaccharides for biomedical applications has been investigated in numerous articles. Alginate has been crosslinked with different divalent cations to produce hydrogels in the form of beads, or as a “single” piece, to deliver drugs and produce scaffolds [25]. Dextran has been used for producing DDSs in the form of particles [16]. In addition, levan has been used for nanoparticles synthesis by taking advantage of its amphiphilic properties [20]. Moreover, some EPSs have special properties for treating different types of cancer because their molecular structure provides them with affinity properties that allow them to interact with different receptors or transporters [28].

As a consequence, several reviews have been published covering and exploring EPSs potential. Hussain et al. reviewed the possibilities of blending different exopolysaccharides to improve their mechanical and biological properties [29]. Specifically, blends with xanthan gum, curdlan, hyaluronic acid, and dextran were discussed, highlighting how their different combinations can be used to obtain versatile materials depending on the final application. For instance, xanthan gum can modify the rheological properties of different solutions, curdlan can confer thermal stability, hyaluronic acid can increase cell adhesion and cell proliferation, and dextran is used as blood plasma substituent. Moreover, the final EPS formulation (hydrogel, particles, etc.) should be also carefully selected depending on the final use. Tabernero et al. [13] reviewed the most common EPSs and their potential for biomedical applications, indicating how this type of polymers can be useful for a high portfolio of applications in biomedicine (cancer therapy, wound healing, nanomedicine, etc.). Again, they pointed out that the polymers must be processed carefully according to the final use. Tchobanian et al. reviewed the potential of some EPSs for tissue engineering [30]. In this case, it was highlighted that although these polymers show a high number of advantages for tissue engineering due to their properties, it is still required to acquire a deeper knowledge concerning interactions between the materials and the cells. That knowledge will serve as a guide to perform a correct material design for any application in tissue engineering. 

These previous reviews focus mainly on some applications of the different polymers, without making reference to the use of EPSs as potential carriers for biomedical drugs. Therefore, the use of EPSs as carriers to develop different biomedical DDSs, including fibers, particles, hydrogels, and aerogels, will be reviewed. Likewise, in vivo experiments that were performed using these polymers as DDSs will be also discussed.

The review is organized in 4 Sections. In Section 2, the DDSs that can be obtained with EPSs are briefly summarized. In Section 3, the different EPSs are explained, with their characteristics and applications, as well their possibilities to produce the previously DDSs. Finally, Section 4 discusses the EPS selection depending on the application, covering their pros and cons and their possibilities for in vivo experiments. 

## 2. Drug Delivery Systems

EPSs can be used to produce DDSs that have many potential biomedical applications. This section summarizes some DDSs that can be obtained and their main applications (Figure 1):

### 2.1. Fibers

Fibers are materials with a high length-thickness ratio. Polymeric fibers can be used as reinforcements for increasing the mechanical properties of a material, or as a surface for attaching and proliferating cells. In this manner, fibers mimic the extracellular matrix; and their use as antibacterial, wound dressing, suture, and scaffold materials has been widely proposed [31,32]. The most common way to produce fibers of different materials is by employing the electrospinning technique and its variants [33].

### 2.2. Gels

A gel is composed by molecules, chains, etc. that are interconnected in a fluid medium thanks to crosslinking, electrostatic interactions, or macroscopic entanglements. Hydrogels are defined as three-dimensional polymeric networks that can absorb a huge amount of water or biological fluids. Gels can be obtained by using crosslinkers or by providing the experimental conditions for the production of physical chain entanglements. Lastly, hydrogels can also mimic the extracellular matrix and be used as a platform for drug release or for cell attachment and proliferation [34,35,36]. 

### 2.3. Aerogels

Aerogels are highly porous materials with a huge specific surface area. Aerogels are obtained after drying a gel under certain conditions that prevent pore collapse. Supercritical CO_2_ (scCO_2_) is used for this drying process (i.e., supercritical drying). Aerogels can be manufactured from different inorganic and organic polymers (e.g., polysaccharides), and due to their structural properties are potential candidates to serve as a cell proliferation platform for different applications. Aerogels can also be loaded with drugs to produce a DDS [37,38].

### 2.4. Exopolysaccharide-Coated Nanoparticles

Nanomaterials are more reactive and effective for any application than any other type of material due to their high surface area/volume ratio. One main nanotechnology application is disease treatment, using a DDS in nanoparticle form. Usually, nanoparticles for DDSs are constituted by an inner “core” material (active compound) and an outer “shell” material (polymer). These can be classified depending on the core or shell material used (e.g., inorganic-inorganic, organic-inorganic, etc.), or on the shell’s or core’s properties [39,40].

Sometimes however, the drug is attached to the surface of the carrier by means of electrostatic interactions or covalent bound. In this case, surface engineering experiments must be performed previously to modify the functional groups of the polymer and to allow a successful drug attachment [41,42]. 

## 3. Exopolysaccharides for Producing DDSs

### 3.1. Alginate

Alginate is a copolymer constituted by β-d-mannuronic and α-l-guluronic acids (molecular weight (0.5–1.5) × 10^6^ Da (Figure 2), which is obtained from the bacterium *Azotobacter vinelandii* [15]. Alginate from bacterial origin and alginate from seawood have similar properties because their structures are also similar. The main difference is that microbial alginate is a stiffer polymer due to the increase in its G-length chain [25]. Alginate is used for different applications, such as wound dressing, as an encapsulating agent, or to increase water solubility. Its gelation, by crosslinking with divalent cations (Ca^2^⁺ or Ba^2^⁺), is well-known in literature and alginate beads have been widely used for different applications [25,43,44].

Alginate gelation technique can be coupled with an electrospinning process to simultaneously induce fiber crosslinking and drug encapsulation. A crosslinking can be a required step since free alginate cannot be processed with this due to its polyelectrolitic behavior. This methodology was used to encapsulate clorhexidine gluconate in 25–30 micron fibers (Figure 3a) with an encapsulation efficiency of around 40%. These fibers had adequate properties to be used for wound healing purposes, including high water uptake and an antimicrobial drug release effect against *Staphylococcus aureus* and *Pseudomonas aeruginosa* (Figure 3b) [45]. The use of alginate for wound dressings is very common in literature since alginate can absorb a high amount of wound fluids as well as provide an ion exchange between calcium and sodium ions. This fact results in a swelling phenomenon that enhances the healing process [46]. These beneficial properties have been also taken into account to produce composite fibers of alginate with poly (vinyl alcohol) (PVA) via electrospinning. This article shows that the production of alginate blends with other polymers via electrospinning is also possible. In this manner, honey was incorporated into the DDS to enhance the inhibition of *Escherichia coli* and *S. aureus* growth [47].

Although alginate hydrogels are mainly produced via a crosslinking phenomenon, thermal gelation and cell-crosslinking are also feasible techniques. Alginate gels can be used as scaffolds for tissue engineering purposes [48]. Growth factors, chondrogenic cells, or even a combination of growth factors, human mesenchymal stem cells, and drugs have also been encapsulated in alginate hydrogels [49,50,51]. 

Alginate hydrogels are also used as fillers and osteoinductive factor carriers for bone engineering. However, their low mechanical resistance is a drawback for this application. This is one of the main reasons why alginate has been blended with other compounds [48]. Moreover, due to their anionic nature, alginate hydrogels can be combined with polymers of cationic nature (i.e., chitosan) to produce different polyelectrolyte complexes [52]. 

Micrometric beads of alginate hydrogels can be easily produced via crosslinking or through dropwise polyelectrolyte formation with an atomization system [53]. These beads are suitable for encapsulating different pulmonary and oral delivery compounds [54], and can even be included in scaffolds to enhance their properties for potential tissue engineering applications [55]. Moreover, alginate’s available functional groups permit its surface to be functionalized, allowing drug attachment [41].

Surface engineering can also be performed to tailor alginate’s final properties. The possibilities are reviewed in [56]. They indicate that, in spite of the limited amount of solvents that solubilize alginate (i.e., a highly hydrophilic compound), this polymer can be modified to attach ligands and even fluorophores used to investigate particle cellular uptake by confocal microscopy.

Alginate aerogels have also been produced by using scCO_2_ drying. The obtained aerogels had high porosity and water uptake values. Their use for wound healing purposes or as a platform for cell proliferation is recommended. 

Alginate has been added to lignin [57] and starch [58] aerogels to increase their hydrophilicity (and water uptake). Alginate-starch aerogels behave as bioactive materials and their use was proposed for bone engineering. Although the addition of starch to these increased their Young modulus, the value is still low (0.5–1.5 MPa) and far from the value for bone [58]. Therefore, their use is better recommended for organs, with a lower mechanical resistance, or wound dressings. 

Aerogel fibers of alginate-chitosan were produced for wound healing purposes. These fibers were non-cytotoxic for fibroblasts and showed proper antibacterial effects towards *S. aureus* and *K. pneumononiae*. Moreover, a scratch recovery of 75% was obtained. This result was similar to that obtained with a conventional medical Kaltostar^®^ device [59]. 

Alginate is perhaps the most frequently used EPS carrier due to its excellent biocompatible and gelation properties. Alginate hydrogels can encapsulate thousands of different compounds, including proteins, growth factors, cells, etc., mainly for pulmonary and oral delivery, tissue engineering, and wound dressing systems. However, their anionic character hinders their fiber production using electrospinning processes. Their stiffness and mechanical properties should be improved by blending alginate with other compounds and polymers or by their chemical functionalization.

### 3.2. Dextran

Dextran is an exopolysaccharide with neutral charge and a molecular weight from 3 × 10^3^–3 × 10^6^ Da (Figure 4). It is obtained mainly from cultivating any strain of *Leuconostoc mesenteroides* [16]. Its main characteristic is the ability to prevent blood clots. It also presents high stability and hydrophilicity properties that can be considered an advantage to improve particle stability and water solubility [60].

Dextran DDSs are usually obtained from its derivatives, such as its oxidized [61] and methacrylated [62] forms, which can be photocrosslinked, by using a suitable crosslinker. 

It is very common to work with dextran blends to enhance the final properties of a DDS depending on the final application. For instance, a methacrylate dextran derivative was blended with gelatin and photocrosslinked to obtain fibers (0.30–1 microns). The authors found a remarkable increase in the blend’s mechanical properties (Young’s modulus 40 MPa) with a water sorption of 1500–2000% in 20 min [63]. Since cells adhered and proliferated in this material, these fibers were considered appropriate for soft tissue engineering due to their mechanical resistance and biological properties [63]. In another example, poly(ε-caprolactone) (PCL) was grafted on to dextran to produce fibers via electrospinning. The addition of PCL resulted useful for producing hydrophobic fibers (the resulting contact angle higher than 100°) and to retard their degradation [64]. Alternatively, dextran can also be attached to fibers of poly(3-hydroxybutyrate-*co*-3-hydroxyvalerate) to confer hydrophilicity and cell proliferation properties [65].

Previous articles have shown how dextran fiber properties can be modified to produce a drug carrier. The drug can be incorporated via a crosslinking methodology or even by electrostatic interactions and π-π stacking, taking into account that the dextran molecule has a large amount of hydroxyl groups. The latter approach was used by Zhou et al. [66] to load paclitaxel (80% efficiency) in fibers. The drug was subsequently released (between 30–40%) depending on the dextran’s molecular weight. This DDS evinced a cytotoxic effect against HeLa cancer cells [66]. 

Dextran hydrogels (mainly dextran derivatives) have been used to encapsulate or attach drugs with different techniques and for different applications, such as tissue engineering. For example, peptides were attached to a methacrylated dextran for regenerating axonal cells [62] and an interpenetrated polymer network (IPN) alginate-methacrylated dextran was loaded with bovine serum albumin (BSA) and chondrocytes for soft tissue engineering [67]. β-tricalciumphosphate has also been loaded in a dextran-based hydrogel, and provided an osteoinductive response according to the bioactivity results [68]. However, the poor mechanical properties of this material hinder its use for bone engineering, and another target application has been considered, namely maxillofacial reconstruction [68]. These articles show the potential use of dextran hydrogels in soft tissue engineering applications, providing the mechanical properties are appropriate. 

Wound healing is another important target for dextran hydrogels, involving neovascularization action or a drug-loaded DDS. Dextran hydrogels can be loaded with particles (i.e., chitosan) that can incorporate growth factors, achieving simultaneously angiogenesis and antifungic properties that stimulate the complete regeneration of the skin without inflammation issues [69].

In spite of the previous facts, dextran has mainly been used to produce nanoparticles for different treatments. Its excellent stability, biocompatibility, and biodegradability (dextran can be degraded with dextranase) have been exploited to coat metal particles, such as iron [70] and manganese/zinc ferrite, for subsequent magnetic resonance imaging or even magnetotherapy [71].

Dextran functional groups have been modified to conjugate any type of molecule to surface particles, enhancing cellular uptake, for example, in cancer cells. This was observed after dextran nanoparticles (600 nm in width and 1 nm in height) were conjugated simultaneously to graphene oxide, an aptamer, and curcumin, using flow cytometry. Doxorubicin was also conjugated to obtain nanoparticles of 100–150 nm for cancer cell treatment [72]. In these cases, it is important to consider that drug release is usually pH-dependent, but that the drug is always released in a controlled manner. Lastly, dextran microparticle blends can also be produced. Microparticles made of a dextran-pullulan blend were loaded with nanohydroxyapatite crystals for filling bone defects (Figure 5a,b). The combination of these polymers confers the nanoparticles osteoconductive and injectable properties, as well as enhances mineralization [73].

Although less extensively studied, dextran has also been used with aerogels. Its coating properties and reactive functional groups were taken into account to coat silica aerogels, attaching 5-fluorouracil to the aerogel’s surface. This system was designed for colon cancer because dextranases can be activated by bacteria in living organisms [74]. 

Dextran is a well-known polysaccharide that can be modified to a large extent due to its biochemical structure. Dextran can be functionalized with many functional groups to encapsulate and/or conjugate hydrophobic and hydrophilic drugs to produce a huge number of drug delivery systems depending on the final application. Moreover, its rheological and injectable properties can be modified depending on the type of crosslinking and derivatives used. 

To conclude, dextran’s bioactivity, hydrophilicity, and stability make this polymer a very promising candidate for coating any type of material. However, as happens with other polysaccharides, its mechanical properties are poor for certain applications and it should be blended with other materials to improve its functionality. 

### 3.3. Xanthan Gum

Xanthan gum (XG) is an anionic polymer obtained from *Xanthomonas campestris* [17] and has a molecular weight of (2–50) × 10^6^ Da (Figure 6). This polymer has different molecular arrangements depending on parameters such as ionic strength and temperature. It is widely used to increase water solubility or viscosity and as a coating agent to stabilize different type of nanoparticles [75,76]. 

XG has a special conformation due to its polymer order–disorder transition: at high temperature (also depending on the ionic strength), XG assumes a coil conformation, whereas at low temperature, the structure is helical [77]. This special phenomenon has been used to produce and load many DDSs (e.g., hydrogel films).

The polymer’s anionic character was taken into account to load blended chitosan-XG fibers with curcumin via electrospinning. A polyelectrolyte complex was formed between both polymers, which presented special properties for intestinal delivery and to treat colorectal cancer. The complex avoids chitosan precipitation in a neutral environment and drug absorption is increased. The obtained fibers had a size of 700–900 nm with an encapsulation efficiency of 70%, and also presented a cytotoxic effect against a colorectal cancer cell line [78]. 

XG has more commonly been used to produce hydrogels by crosslinking with citric acid [79] (Figure 7), or sodium trimetaphosphate [80]. The obtained hydrogels are non-toxic for humans and drug release can be usually controlled, depending on the composition. 

As was mentioned above, XG hydrogel films can be produced by changing the temperature, providing a transition between helical-coil conformations. This strategy is adequate to obtain wound healing systems because XG swells with water and drug release (if encapsulated) may be pH dependent. This hydrophilicity ability of XG is useful to produce blends that increase the swelling and the bioadhesive strength of the final material. There is a wide portfolio of options, such as the possibility of encapsulating amoxicillin with magnetite to produce a material that can release the amoxicillin by means of an electromagnetic field [81]. 

Although rarely done, gums with similar properties to XG can also be processed to obtain aerogels for DDS. For example, the gum tragacanth was used with PVA to confer stability to a subsequent aerogel that was impregnated with a compound with antioxidant activity (silymarin) for oral delivery [82].

The role of XG as a stabilizer, coupled with its reduction power, has been considered frequently for coating metallic nanoparticles. For example, gold nanoparticles were coated with XG and loaded with doxorubicin, or modified with another polymer (i.e., PEG), to incorporate curcumin. The addition of PEG improved an extended systemic circulation of the polymer, and highlights the ability of producing multicomponent systems with a positive combined action. These nanoparticles had an enhanced cytotoxicity activity against lung cancer cells due to XG’s functional groups [83]. 

XG is an EPS with great potential as a DDS because it confers hydrophilicity, viscosity, and adhesiveness to any multicomponent blend. Moreover, its peculiar structure and order–disorder transition provide the polymer with unique properties to obtain hydrogel films, resulting in a high potential EPS for wound healing applications. XG stability has also been taken into account for conferring colloidal stability because of the interactions of its moieties. Moreover, its anionic charge can be used to attach molecules with positive charges. 

### 3.4. Gellan Gum

Gellan gum (GG) is usually obtained from *Sphingomonas paucimobilis* [18] and has a molecular weight ranging from 5 × 10^3^–2 × 10^6^ Da (Figure 8). Gellan Gum is another anionic gum with a peculiar conformation. It is formed by a coaxial double helix that is twisted around the vertical axis. GG presents a structural thermoreversible self-folding phenomenon because temperature controls the polymer’s conformation. This gum is used as a viscosity modifier, stabilizing agent, or to increase temperature stability. Moreover, it can be used to enhance cell proliferation in spite of its poor mechanical properties [27,84,85]. 

GG temperature dependence, as happens with XG, can be used to encapsulate different products in the generated gels. However, its biochemical structure is not the only important parameter concerning GG gel properties. The degree of acetylation of GG plays an important role in the gels mechanical properties, since high temperature stability and more brittle gels result from low-acetylated GG [27,85].

Nevertheless, fibers of GG are difficult to obtain by electrospinning due to its anionic character and strong shear-thinning behavior at low shear rates. The main solution to this situation is to reduce the repulsive forces by blending GG with other polymers such as PVA. Fibers obtained in this manner were hydrophilic, and used for supporting cell proliferation without cytotoxicity drawbacks [86]. As occurs with all EPS fibers, their mechanical resistance can be modified by subsequent crosslinking [87].

GG has been widely used to produce hydrogels for DDSs due to its mucoadhesive properties and ability to promote cell differentiation while maintaining cells at the site of implantation [27,85]. Moreover, GG physical hydrogels can be easily formed by means of a process that involves changing the temperature. When a GG solution in water is heated above 60 °C, the random coil structure predominates, but when the solution is cooled, a double helix is formed, and this order–disorder transition results useful for encapsulating many different substances [27,85].

GG gelation can also be performed with monovalent or divalent cations thanks to the polymer’s functional groups and anionic character. The resulting GG hydrogels have been used for tissue engineering due to their cell differentiation properties. Applications include producing platforms for proliferating cells, skin, bone, or cartilage tissues [88,89,90]. As an example, Figure 9 shows an image with mineralized gellan gum and silk fibroin [90]. In this context, the GG hydrogel surface can be functionalized to produce polymer derivatives that can be important for tailoring a gel’s properties and crosslinking reaction type. The use of methacrylates can be useful to produce, via photocrosslinking, a hydrogel with injectable properties to repair tissue damage [91]. Sulindac and vitamin B12 were encapsulated with this procedure, and the drugs were released between 4–8 h. Moreover, the properties of these injectable hydrogels can also be tuned by adding different compounds, even inorganic clays (i.e., laponite). The clay promoted interactions between the polymer and the clay, increasing the material’s elastic modulus and producing a stronger and more stable gel [92]. However, the use of these hydrogels is limited to certain tissues because of their mechanical properties; although, these can be increased by blending GG with other compounds such as starch [93].

GG has also been used to encapsulate different drugs and materials for different types of delivery. For instance, salbutamol sulfate was encapsulated with GG to treat asthma problems because this increased the drug’s release time and mucoadhesive properties, resulting in an advantage for nasal delivery systems [94]. 

GG (as occurs with XG) may be dried with scCO_2_ to obtain pure polysaccharide (or a hybrid with silica) aerogels [95]. Different types of gelation can be applied previous to a drying process, such as the use of ethanol to increase hydrophobic interactions [96].

GG particles can be obtained using different methodologies. The anionic character of GG allows the possibility of using electrostatic complexation. In this manner, GG reacts with chitosan to encapsulate ketoconazole in a polyelectrolyte complex with antifungal properties against *Aspergillus niger* [97]. Another approach is the functionalization of the polymer to produce a DDS with tailored properties. As an example, GG was modified with polymethacrylamide to encapsulate diclofenac sodium. This surface modification was useful for tuning the drug’s release time [98].

In general, GG is a polymer with a high potential for tissue engineering due to its ability to attach, proliferate, and differentiate cells on its surface. It also presents an order–disorder transition that allows the production of hydrogels with different properties via different methodologies. Moreover, GG is used to produce polyelectrolyte complexes and particles that slow a drug’s release. Nevertheless, GG is mainly used in blends or composites due to its low mechanical resistance and highly polyelectrolyte character; characteristics that hinder its use as the only polysaccharide in fibers and scaffolds. 

### 3.5. Curdlan

Curdlan is a linear β (1→3) glucan with a molecular weight ranging from 5 × 10^4^–2 × 10^6^ Da (Figure 10) that is obtained after cultivating *Alcaligenes faecalis* [19]. This exopolysaccharide has special conformations and different molecular structures depending on the manufacturing process. This polymer can produce irreversible or reversible gels depending on the final temperature; generating a reversible gel when heated up to 60 °C and an irreversible one at 80 °C. Curdlan can be used as a modifier to produce composite scaffolds or wound dressings with enhanced mechanical and biological properties [99,100], but is not soluble in water (although it can be solubilized in water with salt). Finally, this exopolysaccharide has antitumoral properties [101].

Curdlan is an interesting polysaccharide for drug release applications due to its capability to form gels when modifying its temperature. In this fashion, a gel is formed after cooling a previously heated (60 or 80 °C) suspension of curdlan in water (around 5%).

Due their solubility, curdlan fibers have to be obtained by blending them with other polymers such as polyethylene oxide (PEO). For example, tetracycline hydrochloride was encapsulated in these fibers with a diameter of 100–200 nm. These fibers required a subsequent crosslinking because their structure collapsed in water. In any case, this article demonstrated curdlan’s ability to extend the drug’s release time from 5 to 25 h. As well, these nanofibers showed promising antibacterial activity against *Escherichia coli* [102].

Due to its low water solubility, curdlan should be modified chemically, as occurs with other glucans [103], or processed with other polymers before its use in hydrogels or particles. In spite of this drawback, curdlan has many interesting properties for the biomedical field. Curdlan can accelerate the production of cytokines as well as reduce inflammation. For this reason, curdlan was blended with chitosan to produce a transparent hydrogel sheet with antifungal properties for wound healing applications [104]. The resulting hydrogel had enhanced therapeutic effects when compared to two conventional systems. Moreover, the incorporation of curdlan reduced the hydrogel’s wound adherence, which could be an advantage for wound dressing removal. Curdlan has also been blended into other DDSs to control hydrogel swelling (diclofenac encapsulated in a PEO cryogel). However, the use of this glucan confers high levels of hydrophilicity that may not be suitable for growing cells due to surface issues. Moreover, the use of curdlan does not enhance a cryogel’s mechanical resistance, and consequently, authors included cellulose nanofibers to reinforce the structure (Figure 11) [105]. 

The use of curdlan for bone engineering applications has also been proposed. In this case, the polymer was grafted with PVA to improve its mechanical properties. Although the processed scaffold showed an excellent in vivo biocompatibility and was enzymatically degraded (with lysozyme) up to 40% in 28 days, its compressive strength was still lower than 1 kPa [106]. 

Hydrogels of two beta-glucans were dried with scCO2 to produce aerogels for tissue engineering purposes. These aerogels were loaded with acetylsalicylic acid [107] or with dexamethasone [108]. High water uptakes were obtained for both glucans (between 600% and 1400%, depending on the concentration and glucan type). These differences in water uptake promoted a different sustained drug release over 24 h. Since their specific surface areas were always higher than 100 m^2^·g^−1^, this article demonstrates the possibility of producing aerogels via scCO_2_ drying. However, the Young´s modulus of the obtained aerogels was between 8–15 kPa and their application for tissue engineering must be chosen carefully [107,108]. 

Curdlan derivatives are important regarding their future use of this polymer as DDS. Curdlan or β-glucans can be esterified [109], carboxymethylated [110], sulfonylated [111], or even grafted with PEG (to produce an amphiphilic polymer for nanosphere formation) and succinic anhydride (nanoparticle formation by self-assembly) [101]. The latter methodology was followed to conjugate doxorubicin with formed monodisperse nanoparticles (300 nm). At the same time, these nanoparticles were functionalized with polyethyleneimine and the monoclonal antibody trastuzumab to improve cellular uptake and treat Her2+ breast cancer. In vitro and in vivo results showed that this system had a higher antitumoral effect [101]. 

In summary, curdlan, and glucans in general, is a polymer that presents interesting properties for nanoparticle production in cancer therapy and wound healing systems. Its addition, different DDSs can improve their therapeutic effect. However, its low water solubility remains a limitation for its processing, and the use of curdlan for tissue engineering applications should be carefully considered given its poor mechanical resistance. 

### 3.6. Levan

Levan is the only fructose-based EPS (Figure 12) with amphiphilic properties that can be rearranged into nanoparticles (around 150 nm, Figure 13), forming a stable colloidal dispersion when put in contact with water at a concentration higher than the critical aggregation concentration (0.05–0.2 mg·mL^−1^) [24]. Moreover, levan does not form a high viscous solution in water until a concentration about 50% [112] and cannot be gelled without a surface modification. Levan is produced after cultivating bacteria, such as *Zymomonas mobilis* [113] and has a molecular weight around 2000 k9 + Da [24]. This exopolysaccharide shows special antitumoral properties against different cancers due to its GLUT5 recognition ability [114]. Moreover, its addition in composites can increase cell adhesion [115], and it also has adequate properties for cosmetic applications [116].

For these reasons, levan has to be processed with other polysaccharides or functionalized to produce fibers or gels. For example, a sulfated derivative of levan was processed with PCL and PEO using electrospinning technology. The obtained microfibers (1–5 microns in diameter) improved blood clotting thanks to the addition of levan. This work indicated the potential of using levan in blends to produce antithrombogenic compounds [117].

Alternatively, levan was carboxymethylated and subsequently methacrylated to produce a polymer that can be transformed into a hydrogel after its reaction with N-isopropryl acrylamide. The resulting hydrogel was temperature sensitive, and temperature was modified accordingly to control the hydrogel´s swelling ratio and drug release rate (5-aminosalicilyc acid). The addition of levan improved cell viability and retarded drug release [118]. Levan’s ability to increase cell adhesion has also been investigated with other materials, such as membranes [115] or film blends [119]. 

Based on its amphiphilic properties, the main application of levan so far is the production of nanoparticles. BSA [120] and vancomycin [121] have been encapsulated in levan nanoparticles of 200–600 nm with an encapsulation efficiency ranging from 50% to 70%. The protein was released slowly (20% in 10 days), whereas 30% of the vancomycin was released in the first day, maintaining its antimicrobial activity against Gram+ bacteria. 

Another remarkable property of this EPS is its ability to recognize the glucose transporter 5 (GLUT5), which is overexpressed in different cancer cells such as breast and colon. This specificity was studied by Kim et al., who encapsulated a marker (indocyanine green) in levan nanoparticles (100–150 nm) for tumor imaging [114]. In addition, Tabernero et al. attached 5-fluorouracil on levan nanoparticles for colorectal cancer treatment, showing that drug release and particle size was dependent on the pH (maximum release of 70% at 23 h for a pH of 7) [122]. In this context, it is important to specify that the size of levan nanoparticles, obtained from bacterial origin, is strongly dependent on the downstreaming process. In fact, methanol precipitation was shown to be more efficient, avoiding subsequent bacteria proliferation and the precipitation of undesired substances in the medium culture [24].

As occurs with many other EPSs with a high number of hydroxyl functional groups, levan can reduce silver and gold salts to coat metallic nanoparticles. Nanoparticles of silver and gold have been coated before with this polymer to be used as a catalyst [123] or bactericidal system [124]. 

Levan has mainly been used for producing nanoparticles because of its ability to form these by self-assembly (i.e., without surface modification), as well as its specificity with GLUT5. Its potential to improve cell attachment and proliferation has been investigated for different blends by different authors. However, its poor viscosity and solubility in organic solvents does not allow fiber formation without a previous surface modification and makes its direct gelation difficult. 

### 3.7. Hyaluronic Acid

Hyaluronic acid (HA) has a molecular weight of (1–2) × 10^6^ Da (Figure 14) and can be obtained from *Streptococcus equi* [21]. Its potential in the cosmeceutical and pharmaceutical industries is well-known by the scientific community [125]. Hyaluronic acid (HA) is probably the most well-known EPS used in biomedical applications because it is found in the extracellular matrix of different tissues, such as in the vitreous body. In addition, HA is specifically recognized by the CD44 receptor, which is involved in natural cell adhesion properties and overexpressed in many types of cancers. HA is characterized by an expanded random coil structure, which can be entangled to form highly viscoelastic polymeric solutions [125,126,127]. Furthermore, HA is a proper compound for ocular delivery due to its presence in the eye vitreous body [126].

HA can form gels with different crosslinkers. However, previous modification (methacrylate [128] or hydrazide [129]) of the HA is recommended because the crosslinkers are usually toxic and cannot be used to encapsulate cells or other compounds. 

Electrospinning of HA is not produced straightforward. The use of a HA derivative facilitates this process as well as provides the conditions for posterior crosslinking. This approach was followed by Ji et al., [130] to crosslink fibers of thiolated HA with another polymer (poly (ethylene glycol) dyacrylate (PGDA), adding PEO to improve fiber formation via electrospinning. The resulting fibers (100 nm diameter) were non-toxic and permitted fibroblast migration into the material [130]. 

Gelation of HA derivatives has been performed previously to obtain different hydrogels for encapsulating drugs, cells, etc. Some of these were designed as beads for cancer therapies. As an example, photocrosslinked methacrylated HA was used to obtain nanoparticles that were posteriorly loaded with paclitaxel. In this manner, drug release was sustained in a controlled way without an initial burst [131]. Likewise, HA was crosslinked with a PEGylated protein for treating pancreatic cancer. After 14 days, a 100% drug release was obtained, showing activity against pancreatic cancer [132]. Engineering of HA hydrogels has been performed to encapsulate cells and different growth factors, as well as to produce nano- and micro-gels [133]. 

HA has been widely used in solid scaffolds for tissue engineering purposes due to its cell adhesion and proliferation properties, although always presenting problems regarding the scaffolds’ mechanical properties. In this context, graphene oxide was included in a mixture of HA-chitosan in order to increase its mechanical resistance (and biological interaction) for bone engineering purposes. These scaffolds were also loaded with simvastatin (Figure 15) because its release improved the in vitro mineralization of the material. In spite of these results, HA is still mainly used only in hydrogel form for cartilage or nervous system regeneration due to its poor mechanical properties [134].

HA particles have been synthesized via different methodologies. HA’s main application in this form is cancer therapy due to its interaction with the CD44 receptor. For example, HA-carbon dot nanoparticles (100 nm) with Doxorubicin (DOX) were prepared using a hydrothermal method. A drug release between 20% and 40%, depending on the pH, was obtained in 50 h, showing a higher internalization and antitumoral activity compared to free DOX [135]. HA nanoparticles can also self-assemble if the polymer is functionalized properly, as was shown by Wang et al. [136]. These free nanoparticles were internalized more efficiently in the liver according to in vivo imaging results.

Another application for HA particles is ocular drug delivery because HA is found in the vitreous body inside the eyes. This approach was followed by Kalam [137] to coat chitosan-HA gel beads with dexamethasone. Particles of 300 nm were obtained after drying the beads, and these released between 45–70% of the drug in 12 h depending on the particle composition. Moreover, the particles met all the required characteristics in terms of refractive index and transparency for ocular delivery [126]. 

Lastly, HA aerogels have also been proposed for pulmonary delivery. As an example, alginate HA were blended to obtain aerogel beads of 70–100 microns with high porosity (97–98%) and a specific surface area (450–600 m^2^·g^−1^) [138].

HA is perhaps the EPS with the largest variety of applications in biomedicine due to its specific recognition by the CD44 receptor and excellent viscoelastic, cell proliferation, and cell attachment properties. These properties make HA the main EPS for tissue engineering and ocular delivery. However, similar to the other EPSs, chemical modifications are required for its adequate crosslinking and its mechanical properties must be improved by blending it with other polymers. Moreover, HA is obtained by cultivating a *Streptococcus* strain. Consequently, a higher security level is required to obtain this polymer. 

### 3.8. Pullulan

Pullulan is an exopolysaccharide with a molecular weight of (5 × 10^3^–9 × 10^6^) Da (Figure 16) that is obtained after cultivating the fungus, *Aurebasidium pullulans* [22], and is composed by maltotrioses. Pullulan is an EPS with alternate glycosidic bounds that confer thermal stability, high water solubility and adhesive properties, and, in consequence, can be used for producing films [139]. Additionally, pullulan can form flexible fibers because it reduces electric conductivity. Since digestive enzymes cannot process it, oral delivery is another important application when developing drug delivery systems with pullulan [140]. Furthermore, it is recommended as an ideal carrier for liver drug delivery because the lectin receptor in the liver shows affinity for the sugar residues in the pullulan structure. As occurs with dextran or xanthan gum, pullulan can be used for coating metallic particles due to its excellent stabilization properties [141].

Due to the aforementioned properties, pullulan shows great potential to produce fibers and as a DDS for the liver because lectin-like receptors show special affinity for the pullulan sugar residues [142,143]. 

In fact, pullulan has been used in numerous blends to produce fibers via electrospinning. A combination of pullulan/alginate was processed to modify surface tension, viscosity, and entanglement of the mixture, resulting in fibers with different characteristics (Figure 17) [144]. Active substances (e.g., folic acid [145] and certain proteins [146]) were also encapsulated in the pullulan fibers. Electrospinning of proteins is particularly difficult due to their structure; thus, it is particularly important to use pullulan for protein electrospinning. In fact, pullulan’s biochemical structure is advantageous from the electrospinning point of view since it favors interaction with the protein’s functional groups.

As occurs with other polysaccharides with a high number of hydroxyl groups, a previous functionalization [147] or blend with another polymer is required to perform a subsequent pullulan crosslinking for obtaining hydrogels. In this context, it should be specified that DNA can be bound to pullulan after its modification with ethylamines. This system is interesting in that it can deliver DNA intracellularly [148].

Pullulan can also be crosslinked with NaIO_4_ to form an injectable and biocompatible hydrogel with aldehyde groups that can be conjugated with human-like collagen to improve a hydrogel’s mechanical resistance and degradation time [149].

An interesting peculiarity of pullulan, which has been extensively investigated, is the possibility of forming nanogels in water with cholesterol by means of a self-assembly phenomenon that is produced due to the different hydrophobic domains in its structure. These nanohydrogels have been used to entrap different products for different applications, such as prostaglandin E1 for wound healing systems since it promotes neovascularization and wound closure [150]. Different growth factors for bone engineering purposes have also been encapsulated in these particular nanogels [151]. Finally, pullulan has been employed to produce protective films for diverse applications due to its adhesive properties and oxygen barrier [142].

Pullulan has also been considered for particle DDSs, mainly targeting the liver. For example, doxorubicin was conjugated in 50–110 nm pullulan nanoparticles, with a pH sensitive release, for targeting human liver cancer cells with successful results [152]. The use of pullulan for reducing metallic salts to coat silver nanoparticles has also been investigated, achieving antimicrobial action against some fungi and some bacteria [141]. Finally, pullulan can be functionalized to form nanoconjugates with different compounds. For instance, nanoparticles of thiolated pullulan were conjugated with antibodies to remove chromatin fragments and prevent future damage to DNA. In this case pullulan served as the perfect carrier since chromatin degradation normally occurs in the liver [153].

Pullulan is an interesting EPS that presents specific advantages for its use in films and fibers, mainly concerning its structure and viscosity. Its specific affinity with the liver makes this polymer a perfect carrier for liver delivery. Its ability to form nanogels and its stability when coating metallic nanoparticles should also be taken into account for this purpose.

### 3.9. Carrageenan

Carrageenans are anionic polysaccharides with D-galactoses (β (1-4)) as main constituents, and have a molecular weight ranging from (0.1–1) × 10^6^ Da (Figure 18). In this case, they can be obtained from the extracellular matrix of red seaweed (that is why was included as EPS here), such as *Kappaphycus alvarezii* [23]. These polymers have anionic charge due to the existence of sulphate groups. There are different types of carrageenans, but kappa, κ-, iota ι-, and lambda λ-type with one, two, or three sulphate molecules are the most frequently used in biomedical applications. Obviously, the number of sulphate groups modifies the polymer’s properties. In fact, κ- and ι- carrageenans can form thermally-reversible gels that release drugs after thermal stimulation [154]. Another potential application for carragenans is the possibility of attaching drugs to their surface by taking advantage of the structure charge [155]. However, some studies have reported a certain toxicity that hinders its potential applications in biomedicine [156].

Carrageenan has a gel structure at ambient temperature that, coupled with its high hydrophilicity, hinders polymer electrospinning. This inconvenience can be overcome by performing a wet electrospinning with water at 90 °C and salts [157]. The addition of bonds to the resulting fibers (with BaCl_2_ for instance) can serve to align the fibers’ orientation and improve their tensile strength [158]. However, one of the most common strategies is to perform electrospinning of polymeric blends. For example, zein-carrageenan fibers were used to encapsulate rosemary oil and zinc oxide nanoparticles [159]. The addition of the metallic nanoparticles increased the blend’s mechanical resistance, whereas the essential oil conferred additional antimicrobial properties against *S. aureus* and *E. coli*. Nevertheless, one must be careful with a probable cytotoxicity effect that can damage the DNA when the zinc is released. Another interesting possibility that takes advantage of carrageenan’s ionic character is the electrospinning of electrolyte complexes. For instance, quaternized derivatives of chitosan were complexed with kappa-carrageenan and incorporated in fibers with caffeic acid and PHB. The combination of these compounds increased antimicrobial activity against *S. aureus* and *E. coli*, and showed a controlled release of the caffeic acid depending on the fiber composition [160].

Production of carrageenan hydrogels can be based on order–disorder transitions (i.e., cooling a solution of carrageenan), the use of a crosslinker, or even using both phenomena simultaneously. Gelatin is an interesting crosslinker that forms hydrogels with carrageenan after cooling a mixture of both components. This combination provides non-cytotoxic hydrogels with a swelling index of up to 40% and suitable mucoadhesive properties. These hydrogels are also hemo-compatible and can be loaded with ciprofloxacin [161] and quercetin [162]. In this case, drug release was highly dependent on the hydrogel’s composition: 80% of the quercetin was released between 30 min and 15 h, whereas ciprofloxacin release was pH-dependent and showed a similar rate for all investigated conditions at a pH of 7. Carrageenan is another EPS that has been dried with scCO_2_ to produce aerogel beads. Carrageenan beads have been obtained via a gelation dropwise technology with different cations (K⁺, Ca^2^⁺, and Al^3^⁺) [163]. Their texture and size properties were dependent on the type of carrageenan, carrageenan concentration and crosslinkers used. Beads of micrometric size (Figure 19) were obtained with a nanometric pore size and with a specific area from 70–170 m^2^ g^−1^. The best results were obtained using K⁺ as a crosslinker [163]. Moreover, a tubular structure can be formed when using K⁺ as a crosslinker. This tubular structure was investigated to produce aerogel beads for entrapping oils, or oleogels. Results indicated an 80% oil entrapment, demonstrating that this material can be used to entrap lipophilic materials in liquid state [164].

One of the main applications of carrageenan is the production of particles for biomedical applications. Gelation of this polymer with monovalent or bivalent cations produces beads that can serve to encapsulate e.g., ibuprofen (30% drug release in 5 h) and verapamil hydrochloride (70% drug release in 5 h) [165]. 

Carrageenan’s anionic charge can be an advantage to produce nanocomplexes via interactions with cationic polymers such as chitosan. An additional crosslinking can be performed on these polyelectrolyte complexes as Rodrigues et al. [155] demonstrated by crosslinking chitosan with sodium tripolyphosphate. The addition of the crosslinker increased particle stability and reduced particle size from 500 to 200 nm. This article also showed that the initial charge of the different solutions can be modified to control particle characteristics. The idea of producing polyelectrolyte complexes was also explored with proteins, such as protamine (best results with the iota type (100–140 nm)) [166].

Carrageenan is a highly versatile polysaccharide that can be processed to obtain hydrogels following different methods (e.g., order–disorder transitions, monovalent and divalent cations, electrolyte complexes). In this context, it is important to highlight that carrageenan forms a tubular structure with potassium ions, which can be useful for specific applications. Additionally, it is one of the most extensively investigated EPSs for aerogel production due to its straightforward hydrogel formation (also in the form of beads). However, more investigations should be performed to clarify its toxicity.

Finally, as a summary, Table 1 shows the EPS and some of their characteristics. 

## 4. EPS Selection, Rational Design, and in Vivo Results

As was explained in the previous sections, every EPS has its own properties and special functions. Therefore, EPS selection should be dependent on the final application, as Table 2 briefly summarizes.

Cancer therapy is one of the main targets of a DDS. In this case, the selection of the carrier depends on the cancer type and its overexpressed receptors. Breast and colon cancer overexpress the GLUT5 transporter that is associated to fructose. Therefore, levan would be the perfect carrier for targeting this type of cancer cells [114]. Levan’s ability to form nanoparticles is another advantage for developing a DDS of systematic administration. However, if the targeted cancer overexpresses the CD44 receptor, HA is the most proper carrier [135]; whereas pullulan should be chosen for liver cancer [167]. Another promising polymer for cancer therapy applications is curdlan since an improvement in a drug’s therapeutic effect when using this EPS has been shown [101]. 

For wound healing purposes, alginate is perhaps the most adequate polymer because of its ability for swelling and ion exchange [168]. Here, it is relevant to mention the possibilities of developing new types of DDS for wound dressing applications using anionic EPSs to form complexes with chitosan, a polymer with antifungic properties [169].

When the main parameter to be taken into account is the organ to be targeted or the administration methodology, different options can be followed depending on the polymer properties or the size. Pullulan is an appropriate polymer for liver and oral delivery [147,167,170]. EPSs that can form hydrogel microbeads by crosslinking methodologies (i.e., without requiring a surface modification) can be an interesting approach to encapsulate drugs in a permanent network for oral delivery, since a wide particle size distribution is not a problem in this type of delivery. Alginate [25] and carrageenan [171] are great EPS candidates for this application.

Pulmonary delivery requires a lower size than oral delivery. In this case, the use of beads is also recommended; however, the applied technique must be able to achieve particle sizes lower than 5 microns. Atomization systems are a good option for this enterprise [172]. For oral and pulmonary delivery, the use of beads with drug-loaded aerogels (after drying with scCO_2_) allows DDS drying while maintaining the polymeric network. Alginate [38] or its blend with other polymers (even with HA) [138] and carrageenan aerogels [163] have been widely investigated for this purpose. 

Due to its presence in the vitreous eye, HA is the most adequate polymer for ocular delivery [137]. Finally, for intravenous administration, the use of nanoparticles is recommended. Levan [120,121,122] and modified EPSs are other alternatives for producing self-assembling nanosystems (e.g., pullulan-cholesterol [150,151,152] and polyelectrolyte complexes [173]). 

Some articles proposed the possibility of using photocrosslinkable hydrogels in order to obtain injectable materials, indicating that an EPS with a methacrylatable surface should be selected [91,174]. Moreover, weaker, thermally-reversible hydrogels of curdlan and gums can be obtained by physical gelation [79,94]. This phenomenon can be an alternative to produce hydrogels with injectable properties.

Another application for EPSs is tissue engineering. Here, the most appropriate polymer is HA because of its viscoelastic properties and interaction with the CD44 receptor, which is related to cell seeding and differentiation [126]. However, GG also shows excellent properties concerning cell adhesion and proliferation [85,88]. Levan, although mainly used in blends, is another polymer that has been used for improving biological properties concerning cellular adhesion [115,119]. Nevertheless, EPSs should be carefully used for tissue engineering due to their poor mechanical properties. For that reason, they are mainly used in composites for this purpose [29]. 

Finally, the DDS type can be important to select the EPS. EPSs are not usually good candidates for obtaining fibers via electrospinning due to their electrolytic character, and, in other cases, due to their low shear viscosity or lack of entanglement; therefore, different approaches have to be followed to obtain this type of DDS [86,102,117]. Pullulan is the most promising EPSs for producing fibers [140,144,145,146]. The use of EPSs in composite fibers is an advantage that allows to tune their properties, and even enables a subsequent crosslinking that can improve the fibers’ mechanical resistance and prevent their degradation. 

Pullulan is the most promising polymer to produce films [140]. Still, films of other EPSs can be produced using layer-by-layer techniques based on the formation of polyelectrolyte complexes [119,175]. In addition, XG has an inherent ability to produce hydrogel films when changing the temperature due to its order–disorder transition [79]. 

Finally, EPSs with a high number of hydroxyl groups in their structure can be used to reduce metallic salts and coat gold or silver nanoparticles. Dextran [70], levan [124], XG [75], and pullulan [141] have already been used for this purpose.

These results indicate the versatility of EPSs to produce any type of DDS depending on the desired final application. These EPSs have been successfully tested in many in vivo systems. Table 3 shows some experiments with these polymers and the main results obtained. As can be observed, EPSs are ready to be used as materials or blends in biomedical applications.

## 5. Conclusions

The potential uses and biomedical applications of the nine most famous EPSs used as carriers for DDSs have been discussed in this review. Their biocompatibility, non-toxicity, and biodegradability make them attractive materials to carry any type of drug inside the human body. In addition, their structure provides a remarkable architecture that can be transformed to attach proteins, gain ionic character, or produce stimuli response hydrogels. These characteristics supply these polymers with an arsenal of possibilities to produce fibers, hydrogels, films, particles, or aerogels with tailored properties for targeting any type of delivery or organ. However, exopolysaccharides’ mechanical properties should be improved to allow their use in all tissue engineering areas (i.e., bone). Moreover, new techniques that process these polymers should be developed to prevent possible problems with structure or entanglement (i.e., microfluidics to obtain fibers), and more investigations should be carried out to produce aerogels from other polysaccharides aside from alginate.

## Figures and Tables

**Figure 1 polymers-12-02142-f001:**
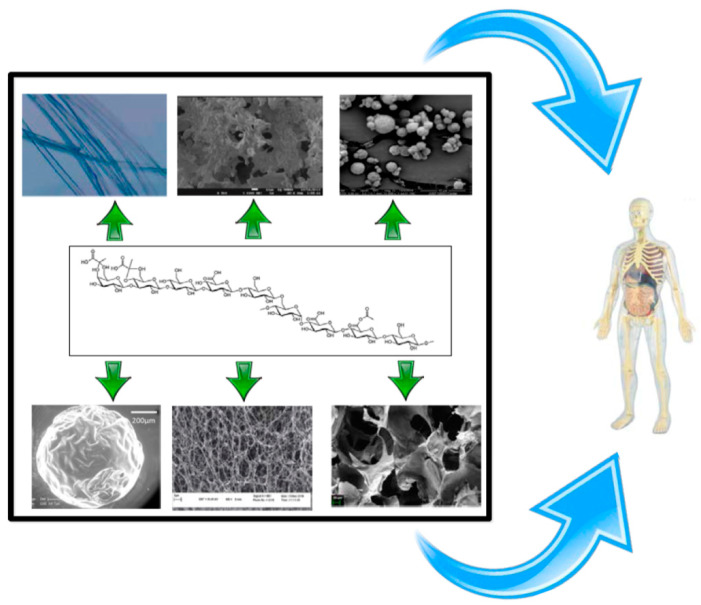
Drug delivery system (DDS) that can be obtained from exopolysaccharides (EPSs) (microparticles, fibers, hydrogel, aerogels).

**Figure 2 polymers-12-02142-f002:**
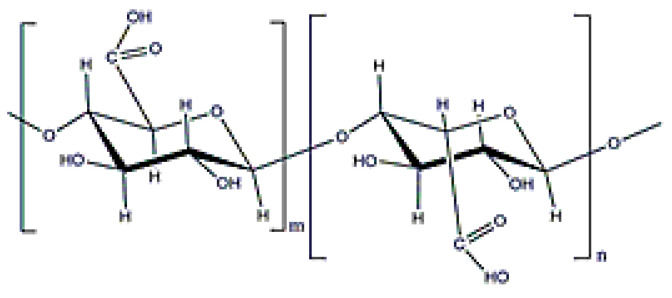
Alginate structure.

**Figure 3 polymers-12-02142-f003:**
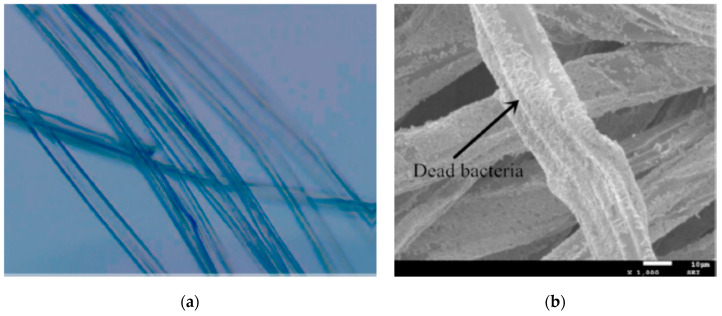
(**a**) Alginate fibers with clorhexidine gluconate; (**b**) dead bacteria (*P. aeruginosa)* on the alginate fibers (adapted with permission from [45]).

**Figure 4 polymers-12-02142-f004:**
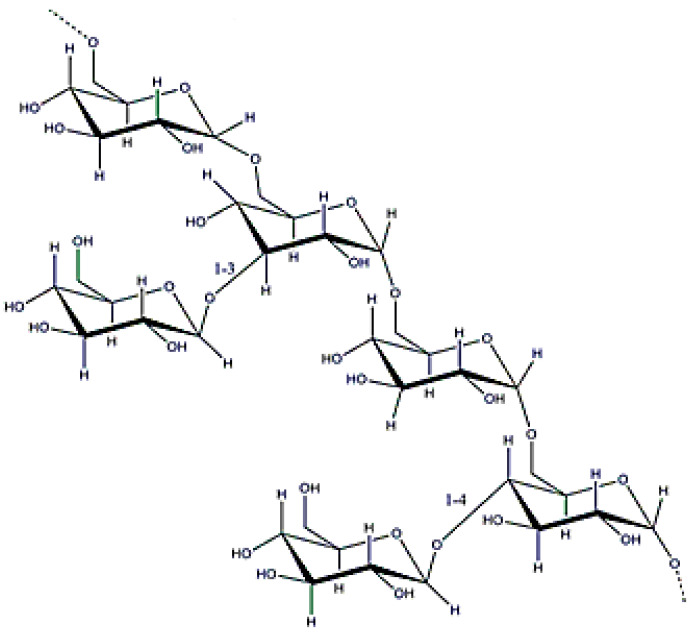
Dextran structure.

**Figure 5 polymers-12-02142-f005:**
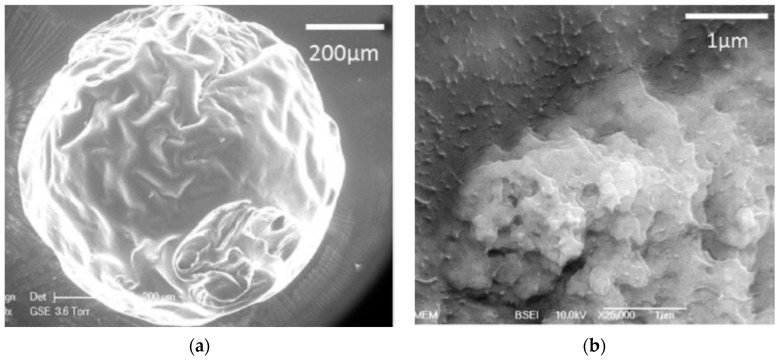
(**a**) Microparticles of pullulan-dextran with nanohydroxyapatite crystals; (**b**) nanohydroxyapatite crystals distribution in the microparticles (adapted with permission from [73]).

**Figure 6 polymers-12-02142-f006:**
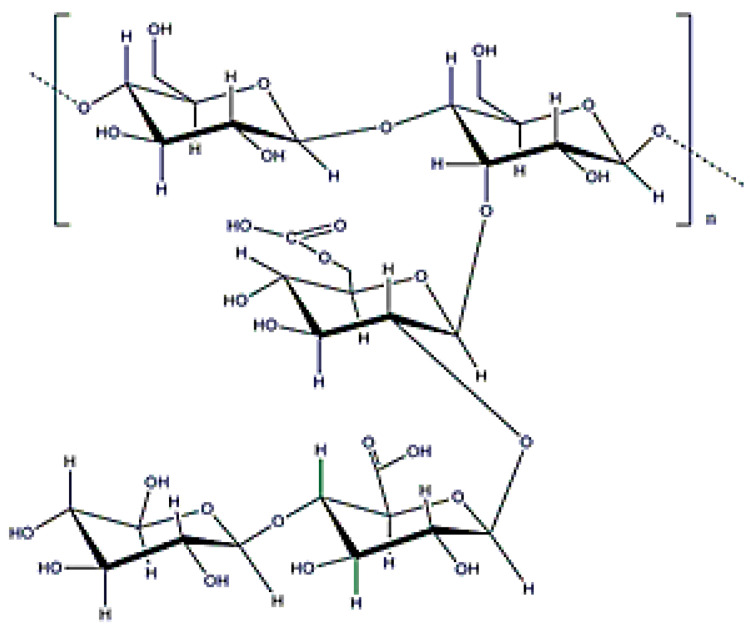
Xanthan gum structure.

**Figure 7 polymers-12-02142-f007:**
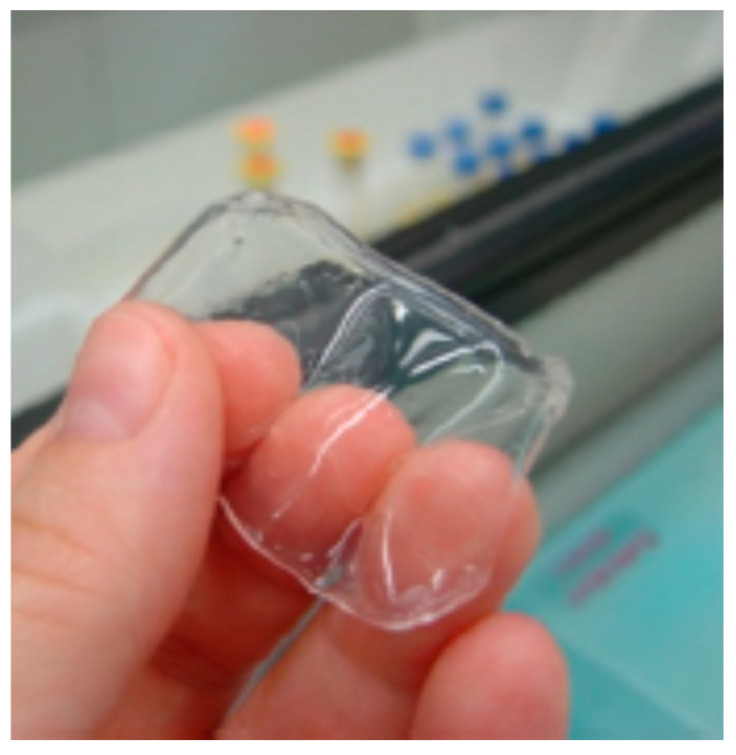
Swollen hydrogel xanthan gum (XG)-citric acid (adapted with permission from [79]).

**Figure 8 polymers-12-02142-f008:**
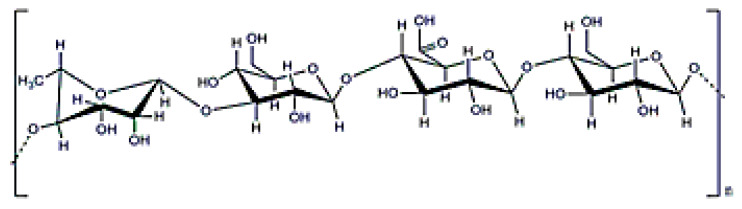
Gellan gum structure.

**Figure 9 polymers-12-02142-f009:**
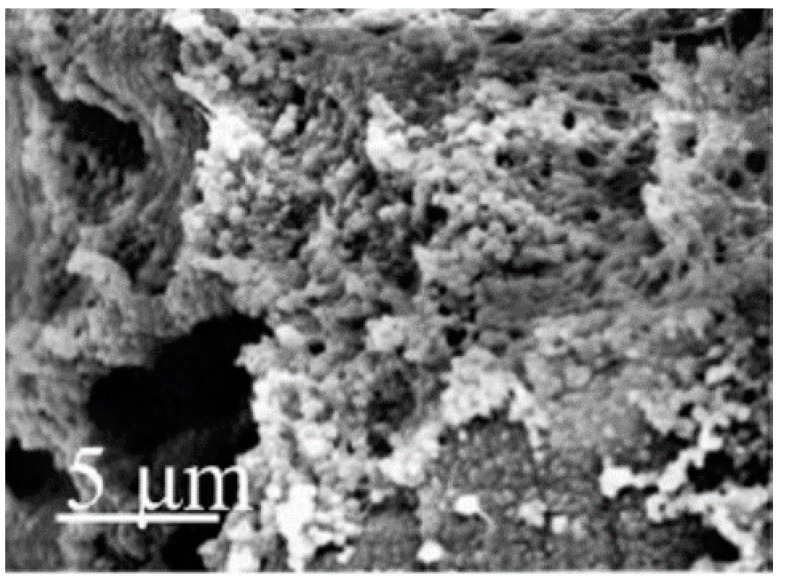
Gellan gum/silk fibroin hydrogel (25% GG and 75% silk fibroin) mineralization (adapted with permission from [90]).

**Figure 10 polymers-12-02142-f010:**
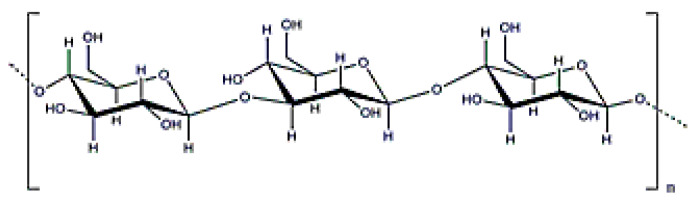
β-d-glucan structure.

**Figure 11 polymers-12-02142-f011:**
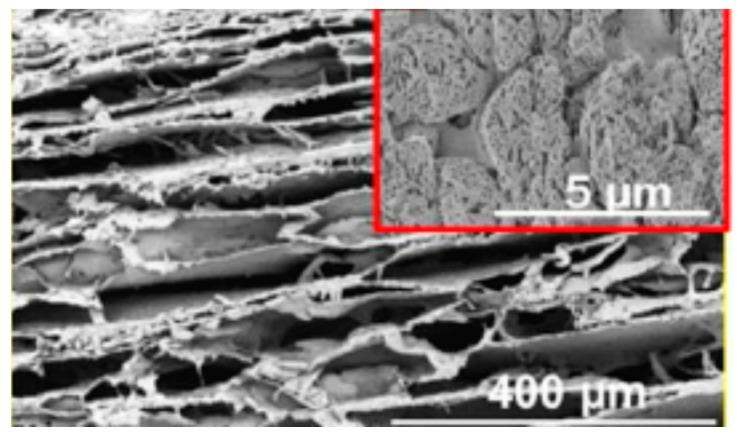
Cryogel of curdlan/PEO (80–20) (adapted with permission from [105]).

**Figure 12 polymers-12-02142-f012:**
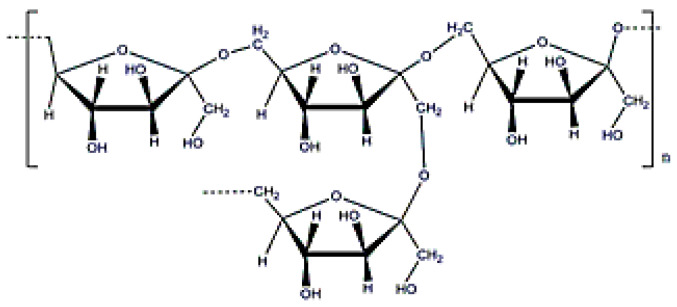
Levan structure.

**Figure 13 polymers-12-02142-f013:**
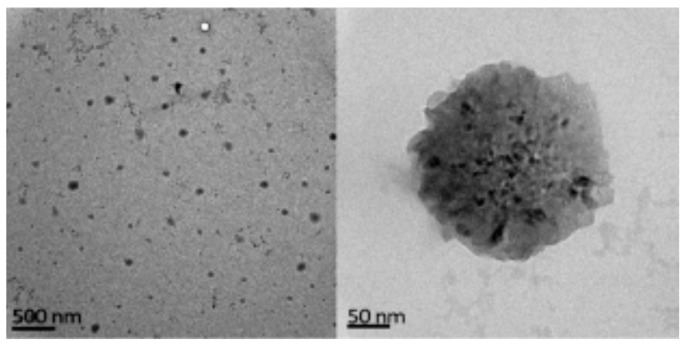
Levan nanoparticles (adapted with permission from [24]).

**Figure 14 polymers-12-02142-f014:**
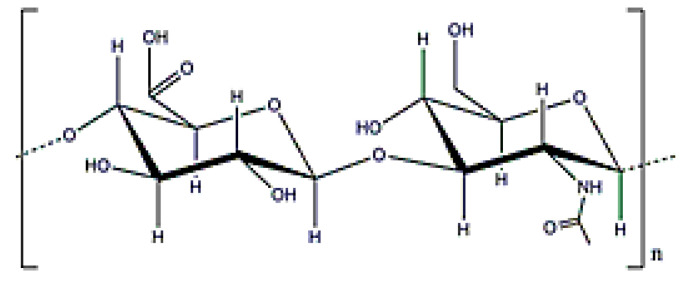
Hyaluronic acid structure.

**Figure 15 polymers-12-02142-f015:**
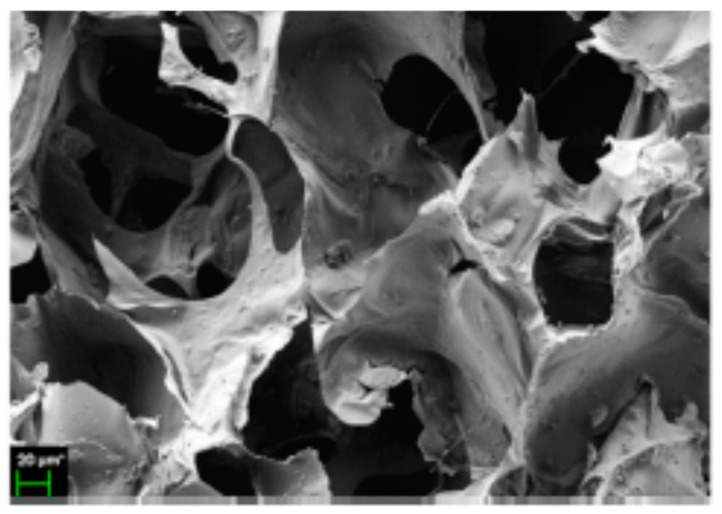
Loaded hyaluronic acid (HA) scaffold with simvastatin (adapted with permission from [134].

**Figure 16 polymers-12-02142-f016:**
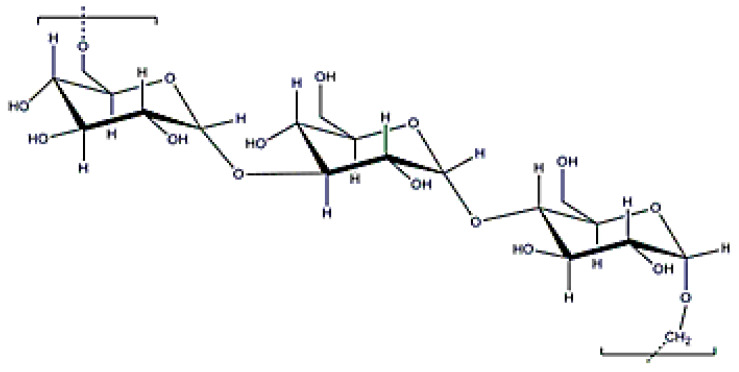
Pullulan structure.

**Figure 17 polymers-12-02142-f017:**
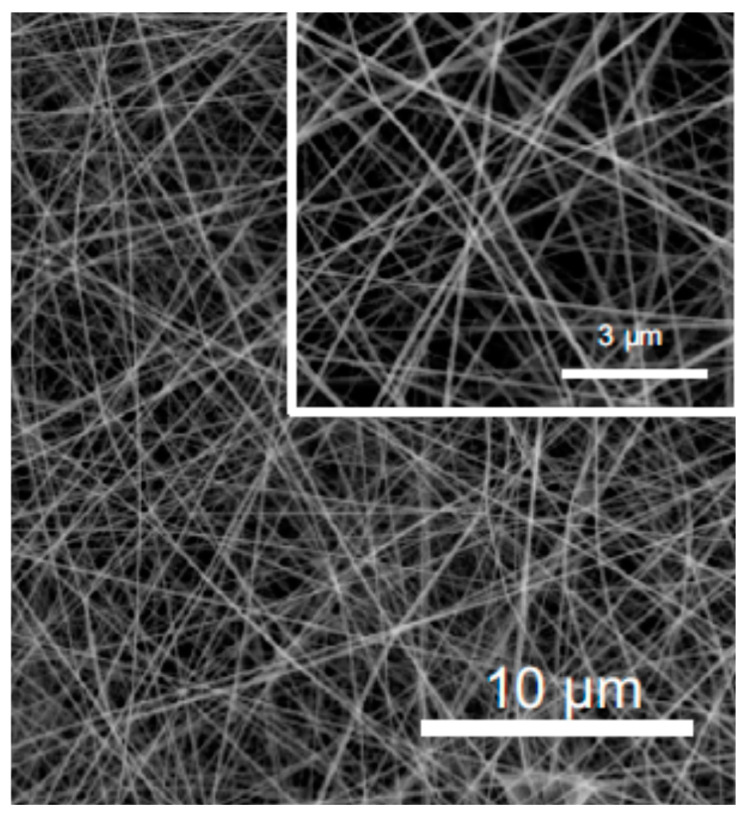
SEM image of pullulan/alginate fibers (adapted with permission from [144]).

**Figure 18 polymers-12-02142-f018:**
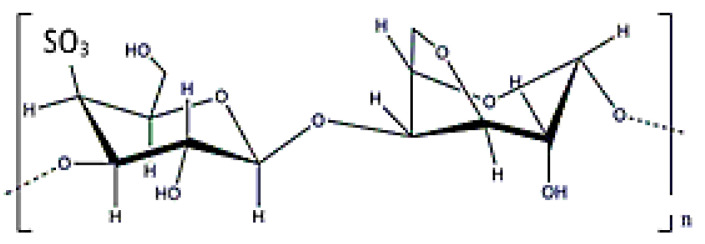
κ-carrageenan structure.

**Figure 19 polymers-12-02142-f019:**
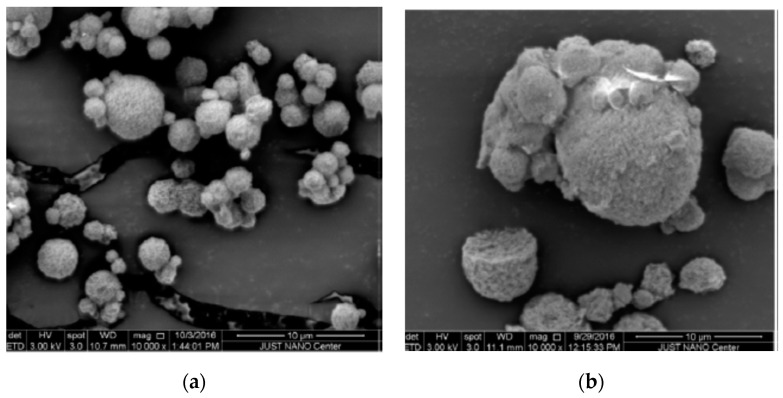
Beads of carrageenan aerogels obtained starting from different concentrations: (**a**) 2% wt. and (**b**) 4% wt. in water (adapted with permission from [163]).

**Table 1 polymers-12-02142-t001:** Some EPSs and some of their characteristics.

EPS	Organism	Molecular Weight (Da)	Structure
Alginate	*Azotobacter vinelandii* [15]	(0.5–1.5) × 10^6^	β-d-mannuronic and α-l-guluronic acids
Dextran	*Leuconostoc mesenteroides* [16]	3 × 10^3^–3 × 10^6^	Branched poly-α-d-glucosides
Xanthan gum	*Xanthomonas campestris* [17]	(2–50) × 10^6^	pentasaccharide repeat units, comprising glucose, mannose, and glucuronic acid
Gellan	*Sphingomonas paucimobilis* [18]	5 × 10^3^–2 × 10^6^	a backbone of repeating unit of β- d-glucose (d-Glc), l-rhamnose (l-Rha), and d-glucuronic acid (d-GlcA) and two acyl groups
Curdlan	*Alcaligenes faecalis* [19]	5 × 10^4^–2 × 10^6^	Linear β-(1,3)-linked glucose residues
Levan	*Zymomonas mobilis* [20]	2 × 10^4^–2 × 10^7^	d-fructo-furanosyl residues joined together by β-(2,6) and β- (2,1) linkages
Hyaluronic acid	*Streptococcus equi* [21]	(1-2) × 10^6^	d-glucuronic acid and *N*-acetyl- d-glucosamine, linked via alternating β-(1→4) and β-(1→3) linkages
Pullulan	*Aurebasidium pullulans* [22]	5 × 10^3^–9 × 10^6^	Maltotriases
Carrageenans	*Kappaphycus alvarezii* [23]	(0.1–1) × 10^6^	d-galactoses (β (1-4)) and sulphate groups

**Table 2 polymers-12-02142-t002:** EPSs with potential applications and some disadvantages.

EPS	Main Applications and Properties	Disadvantages
Alginate	Hydrogel formation and encapsulating agent Particle formation Wound healing	Fibers processing Mechanical properties
Dextran	Blood viscosity reduction Coating agent Easy surface modification due to–OH groups	Mainly in blends Surface modification is required Mechanical properties
Xanthan gum	Stabilize particles Cosmetics Increase viscosity and water solubility Hydrogel film formation	Fibers and aerogels not common Mechanical properties
Gellan	Tissue engineering and bone regeneration Gel formation and high temperature stability	Fibers processing Degree of acetylation changes its properties Mechanical properties
Curdlan	Tumor inhibition Gel formation	Insoluble in water (soluble in alkaline solutions) Mechanical properties
Levan	Encapsulation agent Cosmetics GLUT5 interaction Nanoparticle self-assembly phenomenon Increase cell adhesion Coating agent	Gel formation difficult Not increase in viscosity Mechanical properties
Hyaluronic acid	Cosmetics Tissue engineering and bone regeneration Ocular drug delivery Cancer diagnosis (CD44 interaction) Skin moisturizer	Obtained from *Streptoccocus* Surface modification is recommended for gel forming Mechanical properties
Pullulan	Liver drug delivery Fibers Coating agent Film properties Nanogels Increase water solubility	Aerogels not common Not a significant increase in cell adhesion for bone engineering Mechanical properties
Carrageenans	Carrageenan derivatives with some biological properties of interest Drug delivery by different gelation processes Increase water solubility and gel formation (depending on the carrageenan type)	Obtained from macroalgae Toxicity not clear Properties depending on the carrageenan type Mechanical properties

**Table 3 polymers-12-02142-t003:** EPS examples with applications and in vivo experiments.

EPS	DDS	Blend/Encapsulation	Application	Animal Model	Ref
Alginate	Scaffold	Octacalcium phosphate	Bone regeneration	Mouse	[176]
Dextran	Hydrogel	Microparticles with growth factors loaded in the hydrogel	Wound healing	Wistar rats	[69]
XG	Scaffold	GG and HA	Tendon regeneration	Sprague-dawley rats	[177]
GG	Scaffold	Chitosan and ondansetron hydrochloride	Nasal administration	Rabbits	[178]
Curdlan	Nanoparticles	Doxorubicin/trastuzumab/Polyethylenimine	Cancer treatment	Mice	[101]
Levan	Nanoparticles	Indocyanine green	Breast cancer imaging	Mice	[114]
HA	Nanoparticles	Carbon dox and doxorubicin	Breast cancer therapy	Mice	[135]
Pullulan	Nanogels	Cholesterol-bearing pullulan with prostaglandin E1	Wound healing	Wistar rats	[150]
κ-carrageenan	Hydrogel		Biocompatibility studies for regenerative medicine	Rats	[179]

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
