# Peer review of "Microbial Exopolysaccharides as Drug Carriers"

_polymers, 2020, doi:10.3390/polym12092142_

Round 1

Reviewer 1 Report

This manuscript entitled “Microbial exopolysaccharides as drug carriers” discussed the potential use of exopolysaccharides as carriers for drug delivery systems, covering their versatility and their vast possibilities to produce particles, fibers, scaffolds, hydrogels, and aerogels with different strategies and methodologies. Moreover, the main properties of exopolysaccharides were also explained, providing information to achieve an adequate carrier selection depending on the final application. I think this manuscript can be accepted for publication in Polymers after addressing several minor points as following:

  1. Some necessary references were missing and should be added, for example, in the first paragraph of section 1: “One of the most important parameters for a DDS is its size because it determines the organ that can be targeted…”.
  2. A superficial list of what have been done is not enough and the authors should give more insight into what the outcomes were of the published work.
  3. It is recommended to replace the figure in the text with high-definition figures, especially Figure 5, Figure 6, and Figure 10.
  4. The characteristics of microbial exopolysaccharides have been introduced in detail in Section 2, and the brief introduction at the beginning of Section5.1, Section 3.5.2, Section 3.5.3, Section 3.5.4, Section 3.5.5, Section 3.5.6, Section 3.5.7, Section 3.5.8, Section 3.5.9 are slightly repeated.
  5. It is recommended that the author check whether the references in the text are quoted correctly, for example the end of the first paragraph of 3.5.4(i.e. [19-85]).
  6. The authors could consider adding the following review articles and references which would again increase the interest to general biomaterial readers: Journal of Controlled Release 2018,273, 160-179;Materials Horizons, 2020, 7, 746-761;CCS Chemistry, 2020, 2, 128-138.

Author Response

Dear Reviewer,

We would like to thank the reviewers for their helpful comments. In the following lines, we specify all corrections and changes in the manuscript and we adressed all issues raised by the reviewers one by one.

REVIEWER 1

This manuscript entitled “Microbial exopolysaccharides as drug carriers” discussed the potential use of exopolysaccharides as carriers for drug delivery systems, covering their versatility and their vast possibilities to produce particles, fibers, scaffolds, hydrogels, and aerogels with different strategies and methodologies. Moreover, the main properties of exopolysaccharides were also explained, providing information to achieve an adequate carrier selection depending on the final application. I think this manuscript can be accepted for publication in Polymers after addressing several minor points as following:

  1. Some necessary references were missing and should be added, for example, in the first paragraph of section 1: “One of the most important parameters for a DDS is its size because it determines the organ that can be targeted…”.

Corrected. We added around 30 new references in the manuscript.

  1. A superficial list of what have been done is not enough and the authors should give more insight into what the outcomes were of the published work.

Done. We added some paragraphs to explain the main conclusions of the published works.

  1. It is recommended to replace the figure in the text with high-definition figures, especially Figure 5, Figure 6, and Figure 10.

Done. We hope that now the new figures satisfy reviewer´s requirements.

  1. The characteristics of microbial exopolysaccharides have been introduced in detail in Section 2, and the brief introduction at the beginning of Section5.1, Section 3.5.2, Section 3.5.3, Section 3.5.4, Section 3.5.5, Section 3.5.6, Section 3.5.7, Section 3.5.8, Section 3.5.9 are slightly repeated.

The reviewer is completely right and reviewer 3 made a similar comment. In order to not repeat the same concepts, we removed the section 2, and we introduced the microbial exopolysaccharides in section 3, before the applications as drug carriers. In this way, we can satisfy both reviewer 1 and reviewer 3 requirements.

  1. It is recommended that the author check whether the references in the text are quoted correctly, for example the end of the first paragraph of 3.5.4(i.e. [19-85]).

Thanks for the indication. Done.

  1. The authors could consider adding the following review articles and references which would again increase the interest to general biomaterial readers: Journal of Controlled Release 2018,273, 160-179;Materials Horizons, 2020, 7, 746-761;CCS Chemistry, 2020, 2, 128-138.

We thank the reviewer for these references. These references are now included in the introduction.

Reviewer 2 Report

The title of the manuscript by Antonio Tabernero and Stefano Cardea, i.e. “Microbial exopolysaccharides as drug carriers” makes the reader think that the review should provide an updated overview regarding the properties and potential use of biofilm exopolysaccharides produced by bacteria, as carriers for drug delivery systems.

In the abstract, the authors assert that these polymers specifically secreted by bacteria to a medium culture during the biofilm formation, and where they can be subsequently recovered, are biocompatible and biodegradable, possess specific and beneficial properties for biomedical drug delivery systems, can have antitumor activity, can produce hydrogels with different characteristics due to their molecular structure and functional groups, and they can even produce nanoparticles via a self-assembly phenomenon.

Although all assertions appear very appealing, unfortunately, along the manuscript they are not confirmed by reported case studies, referring specifically to exopolysaccharides recovered by bacterial biofilm. Differently, the review, that only minimally reports the available studies on the use of bacterial exopolysaccharides as drug delivery systems (and not necessarily obtained by bacterial biofilm, but for example by the fermentative action of bacteria on natural starch, as Xanthan gum, Ref. 18), reviews mainly the case studies regarding the properties and the potentialities of exopolysaccharides differently obtainable. In Introduction section, a brief part of the manuscript really deals with bacterial exopolysaccharides, as constituents of the bacterial biofilm, with how and why they are produced by bacteria and with how to produce them industrially. In this regard, the microbiological correctness of the information given by the authors, regarding the biofilm is highly questionable or at least confusing.

The same authors, at the end of Introduction, highlight the relevance and novelty of their review in comparison to already existing ones regarding exopolysaccharides as materials with potential for biomedical applications, saying that “several reviews have been published covering the different strategies to process EPSs and explore their potential” but that “These reviews focus mainly on the general applications of different polymers, without making reference to the use of microbial exopolysaccharides as potential carriers for biomedical drugs”.

Moreover, the authors conclude the Introduction section asserting that in the present review, they have covered the use of bacterial EPSs summarized in Section 2 as materials to develop different types of DDSs including fibers, particles, hydrogels, and aerogels and their applications (Section 3) and finally that they have dedicated Sections 4 and 5 to discuss the use of bacterial EPSs for DDSs, covering their pros and cons and their possibilities for in vivo experiments.  Unfortunately, this announced scenario does not find an actual development in the manuscript.

  • First of all, Section 5 encompasses the Conclusions, and therefore it is not a section dealing with a discussion on the possibilities of bacterial exopolysaccharides for in vivo experiments.
  • As for Conclusions, the authors assert to have discussed “the potential uses and biomedical applications of the nine most famous EPSs used as carriers for DDSs”, and it is true, but none of them has been actually obtained by treating bacterial biofilms, as confirmed by the cited references (150, 72, 151, 152, 23, 24, 124, 136, 153). From the title, Abstract and Introduction, the scope of the work had to be to review the uses and properties of exopolysaccharides produced by bacteria during biofilm formation, but it has not been addressed.
  • Section 2 actually discusses the main types of exopolysaccharides achievable by bacteria cultures, but with exception for Ref. 8 (a chapter from the same author of the present work, Tabernero, that deals with bacterial exopolysaccharides, and from which Figure 2 derives), and Ref. 11 on bacterial dextrane, no reference exists confirming the bacterial source reported for each polymer described. In addition, the reported references concern the properties and uses of exopolysaccharides not obtained by bacterial biofilm.

The same criticisms and the same inconsistencies between the aims and contents of this review can also be found in Sections 3 and 4. Based on these general considerations, this review is absolutely not eligible for the publication on a respectable journal such as Polymers.  

Furtheremore, if we want to go into detail, here are some problems encountered step by step in reading this work.

  • Lines 21-23. The keywords are missing
  • Line 26 “half-life”, in my opinion, is better than “time of activity”
  • Lines 31-32 “it is a pivotal factor for targeting the organs”, in my opinion, is better than “it determines the organ that can be targeted”
  • Line 38-39, a DDS provide additional advantages to protection of drugs from degradation. The authors have given limited information in regard.
  • Line 46. PLLA and PLGA need of the complete names when cited for the first time
  • Line 46. If the authors say “by many researchers”, one Ref. is not sufficient
  • Line 49. “That usually are linked by glycosidic bonds”, in my opinion is better than “ that usually bind by glycosidic linkages”
  • Line 49. Better if the beginning of the sentence is removed. Please start with Polysaccharides are classified…”
  • Lines 52-67. Not clear. The description of the biofilm formation and of its components, including EPSs, must be improved and the rest of sentences have to be reformulated.
  • Line 100. Change the Figure caption in Molecular structure of (microbial?) exopolysaccharides.
  • In my opinion, the text of Section 2, might be removed and replaced with a Table.
  • Line 176. I suggest to create Section 3.2. Gels, and then to divide in subheadings as 3.2.1. Hydrogels and 3.2.2. Aerogels.
  • Also for Section 3, in addition to the text, I suggest to create a Table summarizing the information and to shorten the main text.
  • When there are references whose numbers are consecutive the hyphen should not be used to separate them but the comma. E.g. line109, [10, 15-16] is not correct. The right form is [10, 15, 16]. The error is repeated several times throughout the manuscript.

And I don't go any further.   

Author Response

Dear Reviewer,

Reviewer 3 Report

This review summarizes the potential use of exopolysaccharides as carriers for drug delivery systems, covering their possibilities to produce particles, fibers, scaffolds, hydrogels, and aerogels with different strategies and methodologies. This review also explains the main properties of exopolysaccharides, which provides information for selecting suitable carriers according to the final application. Here are some suggestions for the improvement of the review.

  1. The structure of this review is not clear enough. There is no need to introduce the exopolysaccharides separately in Section 2, which can be added in 3.5.Section 3.1-3.4 are not necessary in this review and can be adjusted to other parts. In a word, the structure of this review should be carefully adjusted before published.
  2. Several grammatical mistakes should be modified. E.g. “cannot be electrospun” in line 208.
  3. Some expressions are confused. E.g. “This article” in line 216.
  4. References need to be noted in the text of Section 4.

Author Response

Dear Reviewer,

We would like to thank the reviewers for their helpful comments. In the following lines, we specify all corrections and changes in the manuscript and we adressed all issues raised by the reviewers one by one.

REVIEWER 3

This review summarizes the potential use of exopolysaccharides as carriers for drug delivery systems, covering their possibilities to produce particles, fibers, scaffolds, hydrogels, and aerogels with different strategies and methodologies. This review also explains the main properties of exopolysaccharides, which provides information for selecting suitable carriers according to the final application. Here are some suggestions for the improvement of the review.

  1. The structure of this review is not clear enough. There is no need to introduce the exopolysaccharides separately in Section 2, which can be added in 3.5.Section 3.1-3.4 are not necessary in this review and can be adjusted to other parts. In a word, the structure of this review should be carefully adjusted before published.

We completely agree with the reviewer. In fact the reviewer 1 also commented a similar fact concerning section 2. We removed section 2 and we introduced the EPSs in section 3.5 as the reviewer requires-

Concerning sections 3.1.-3.4 we believe that the best way to not confuse the readers with the different DDS is to introduce their definitions separately. As a consequence, we do not have to adjust that definitions along the manuscript and we can address directly the polymers´ uses as drug carriers.

We kindly ask the reviewer to reconsider this correction. In any case, if the reviewer thinks that we have to remove these sections (3.1-3.4) as a mandatory task, we would be glad to do that.

  1. Several grammatical mistakes should be modified. E.g. “cannot be electrospun” in line 208.

Corrected. We checked the whole manuscript and we correct some expressions. We hope that now the text is OK for the reviewer.

  1. Some expressions are confused. E.g. “This article” in line 216.

Corrected. As in the previous comments, we hope that now the text is OK for the reviewer.

  1. References need to be noted in the text of Section 4.

We thank the reviewer for this correction. We added around 30 new references, covering section 4 and more parts of the manuscript.

Round 2

Reviewer 2 Report

Although the authors have extensively revised their manuscript, they have absolutely not satisfied the requests to organize some parts of their work in Tables that would make the information clearer and more immediate. In my opinion, Tables are a fundamental element in a review. Furthermore, the answer to the point concerning line 100 is not clear. What table are the authors talking about? Also regarding my comments on the references, I think that the authors have not correctly interpreted neither the Polymers guidelines nor my comment. [1,3] means reference 1 and reference 3, while [1-3] means references from 1 to 3, or 1,2,3. The hyphen should therefore not be used when indicating references with consecutive numbers such as 15 and 16 [15,16] or isolated references, such as 15 then 18 then 20 will be [15,18,20] and not [15-18-20] which would like say 15,16,17,18,19 and 20.

The review must be furtherly improved. It is not suitable for publication in the present form.

Author Response

Dear Reviewer,

Reviewer 3 Report

The authors have successfully addressed this reviewer’s comments. It is suggested to be published.

Author Response

Thank you

Round 3

Reviewer 2 Report

The authors have sattisfyied my request. The manuscript can be pubblished, now.